# How Socioeconomic, Health Seeking Behaviours, and Educational Factors Are Affecting the Knowledge and Use of Antibiotics in Four Different Cities in Asia

**DOI:** 10.3390/antibiotics10121522

**Published:** 2021-12-13

**Authors:** Susan Ka Yee Chow, Xingjuan Tao, Xuejiao Zhu, Atsadaporn Niyomyart, Edward Choi

**Affiliations:** 1School of Nursing, Tung Wah College, Hong Kong, China; 2School of Nursing, Shanghai Jiao Tong University, Shanghai 200025, China; tao.xingjuan@shsmu.edu.cn; 3School of Nursing, Hangzhou Normal University, Hangzhou 311121, China; jj_ice@163.com; 4Ramathibodi School of Nursing, Mahidol University, Bangkok 10400, Thailand; atsadaporn.niy@mahidol.ac.th; 5Department of Anaesthesiology, The University of Hong Kong, Hong Kong, China; edwardkyc@gmail.com

**Keywords:** antibiotic use, antibiotic resistance, socioeconomic, education, policies, Asian cities

## Abstract

Antibiotic resistance is occurring widely throughout the world and is affecting people of all ages. Socioeconomic factors, education, use of antibiotics, knowledge of antibiotics, and antibiotic resistance were assessed in four cities in Asia, namely Hong Kong, Shanghai, Hangzhou, and Bangkok. A survey using cluster sampling was used in 2021 to collect data on 642 subjects. Hongkongers used less antibiotics and were knowledgeable about using antibiotics to treat diseases, while Shanghainese were knowledgeable about antibiotic resistance. The multi-linear regression model reported that respondents who lived in Hong Kong (β = 0.744 (95% CI: 0.36–1.128), Shanghai (β = 1.65 (95% CI: 1.267–2.032), and Hangzhou (β = 1.393 (95% CI: 0.011–1.775) (reference group: Bangkok), who had higher scores on antibiotics knowledge (β = 0.161 (95% CI: 0.112–0.21)), higher educational attainment (β = 0.46 (95% CI: 0.296–0.625)), and who were more likely to consult a doctor on using antibiotics (β = 1.102 (95% CI: 0.606–1.598)), were more likely to give correct answers about antibiotic resistance, *p* < 0.001. Older respondents were less likely to answer the items correctly (β = −0.194 (95% CI: −0.333–−0.055), *p* < 0.01. When educating the public on the proper use of antibiotics and antibiotic resistance, multiple strategies could be considered for people from all walks of life, as well as target different age groups.

## 1. Introduction

In 2015, the World Health Organization (WHO) reported that antibiotic resistance is occurring everywhere in the world and is affecting people of all ages. [1]. Each year, antibiotic resistance kills an estimated 700,000 people worldwide [2], while self-medication with antibiotics promotes antibiotic resistance [3]. There is a crucial need to increase people’s knowledge and promote the rational use of antibiotics in humans.

China is a country where a large number of respondents reported that they had obtained antibiotics from a friend or family member [1]. A cross-sectional study conducted further revealed the serious misconception that antibiotics are effective against viral infections [4]. For children, an even higher rate of misuse of antibiotics has been reported, especially in connection with diarrhoea. The protective factors against the misuse of antibiotics are being a female child, having guardians with higher education, and possessing basic health knowledge [5]. Additionally, in China, having an employer sponsor a mother’s medical fees, the mother’s level of education, and the severity of one’s illness have been found to be significant predictors of self-medication [6].

In 2011, a survey revealed that Hong Kong residents who had a low level of education and were over 60 years of age had inadequate knowledge of self-medication; with a majority believing that chronic diseases should be self-managed [7]. Likewise, a focus group interview showed that most respondents were unsure of the causes and nature of antibiotic resistance. [8]. A large-scale household survey in 2017 showed that only a small percentage of the respondents had heard of antimicrobial resistance (36.8%), while a large percentage wanted the decision to take antibiotics to be one arrived at with their input, after a discussion with their doctor [9].

In Thailand, respondents were recently shown to be frequently unable to distinguish antibiotics from other medicines [10]. During interviews with key informants, it was found that Thailand imports antibiotics and that there is no system to monitor the distribution of the active ingredients, with most antibiotics being purchasable for home use without a prescription [11]. Recently, Thailand was the only member state in the WHO’s Southeast Asia Region to have set a target for improving the knowledge and awareness of its citizens about antibiotic resistance.

In Sweden, a high-income country within the North European region, 80.7% of the respondents agreed that bacteria could become resistant to antibiotics, only 19.1% had the misconception that common colds could be cured more easily using antibiotics, and most patients trusted doctors to not prescribe antibiotics [12]. In Denmark, 97% of the antibiotics that were used had been obtained after a medical consultation with a doctor [13]. By contrast, in Kuwait, the second-richest country in the Arab Gulf States, less than half of the respondents had adequate knowledge about antibiotic use and antimicrobial resistance [14]. A systematic review and meta-analysis of 34 studies on household antimicrobial self-medication conducted in low and middle-income countries in Sub-Saharan Africa, Asia, the Middle East, and South America revealed that past successful use, the female gender, older age, and middle income were associated with self-medication [15].

Regarding the determinants of self-medication with antibiotics in Anglo-Saxon European countries, a systematic review reported that these included position at work, employment status, and patient knowledge [16]. Another systematic review in 2019 found that other than patient knowledge and treatment characteristics, doctor–patient interactions led to better adherence to the prescribed regimen [17]. These published studies showed that the prevalence of self-medication with antibiotics is related to education, socioeconomic status, and doctor–patient relationship. Although the determinants of inappropriate use are complex, the suggested solutions have mainly been confined to community education and the enforcement of regulations on antibiotics. Other solutions have included enhancing doctor–patient communications and dispensing medications in exact numbers of doses to avoid leftover antibiotics [18,19].

Current evidence points out that socioeconomic, policies, and educational factors are known to play an important role in illness behaviours and antibiotics consumption. The profound differences between Eastern and Western cultures are manifested in their political systems, histories, and religions. The findings from previous studies conducted in Western countries cannot be extrapolated to Eastern cultures. The present study is designed to fill this gap in knowledge and to provide a snapshot of the current situation on the prevalence of antibiotic use in Asian countries. It is not only providing an estimate of the magnitude of the problem, but would also allow for an evaluation of the health promotion programmes on antibiotics. The main question in our study is whether demographics and city of residence can predict the knowledge and use of antibiotics in Asian countries.

## 2. Results

A total of 680 participants were reached in our study. After checking the questionnaires, 38 questionnaires were discarded because more than 10% of the items were found to be invalid. Those items were related to participants choosing more than one option or having nil replies for the questions. A total of 642 valid questionnaires were collected from the four cities. Among the valid questionnaires, 161 were from Hong Kong, 160 from Shanghai, 160 from Hangzhou, and 161 from Bangkok. The percentage of valid questionnaires was 94.4%.

### 2.1. Demographics Characteristics

With regard to gender, the proportion of males in Shanghai was 65.6%, significantly higher than in Hong Kong (46%) and Bangkok (42.2%), with *p* < 0.001. In terms of age, 53.4% of the participants in Hong Kong were 18-30, compared to 22.5–33.8% in the other three cities. Among those in the age group of 41–55, the proportion in Hangzhou was 43.8%, compared to 12.4–23.1% in the other three cities, with *p* < 0.001. With regard to educational attainment, 75% of the respondents in Shanghai had a sub-degree or bachelor’s degree, compared to 44.7–57.1% in the other cities, with *p* < 0.001. In terms of marital status, 71.9% of the respondents in Hangzhou were married with children, compared to 25.5–44.1% in the other cities. Moreover, 63.4% of Hongkongers and 47.8% of Thais were never married, compared to 19.4–20.0% of their counterparts, with *p* < 0.001. For occupation, 38.8% of the Shanghainese were professionals, compared to a lesser percentage in the other three cities. Regarding economic situation, narrative descriptions were used to indicate the current financial situation of the respondents. They are used to examine subjective feeling of the participants about their current economic situation. The descriptors were from very adequate to very inadequate, as non-local readers would find it difficult to interpret monetary amounts. Thailand had the highest proportion of respondents who claimed that their economic situation was adequate, at 78.4%, compared to 23.8–42.2% of the respondents from the other cities. Half of the respondents from Hangzhou claimed that their economic situation was barely adequate, compared to 31.7–41.9% of their counterparts, with *p* < 0.001. With regard to religion, the majority (93.8%) of Thais were Buddhist, while 21% of the respondents from Hangzhou indicated that they were Christian. As for the number of household members, the median was 4 in Hangzhou and 3 in the other cities, with *p* < 0.001. Please refer to Table 1 for details.

### 2.2. Use of Antibiotics

With regard to the use of antibiotics, nearly half of the respondents in Hangzhou and Shanghai had taken antibiotics in the past 6 months, compared with the Hongkongers and Thais who were using less antibiotics, with *p* < 0.001. As for the sources of the antibiotics that they last took, 72.3% of Thais got their antibiotics from a medical store or pharmacy, compared with 29.4–31.8% of the respondents from the other three cities. Only 8.5% of Thais got their antibiotics from a hospital or clinic, *p* = 0.003. Regarding whether the respondents preferred to consult a doctor who had declared that antibiotics should be used responsibly, 85.7% of Thais gave a positive response compared to less positive responses from the other respondents, with *p* < 0.001. When the respondents were asked whether they had brought a child aged 15 years old or below to consult a doctor in the past 12 months and asked for antibiotics, 70.6% of Thais responded in the affirmative, compared to 16.7–38.6% of the other respondents, with *p* < 0.001. Please refer to Table 2 for details.

### 2.3. Knowledge about Using Antibiotics for Treating Diseases

The participants were asked which of 12 medical conditions could be treated with antibiotics. Among the list of diseases, only wound infections, gonorrhoea, and urinary tract infections can be treated with antibiotics. The median for items that were answered correctly by Hongkongers was 7, with an interquartile range of 4–9, compared to a median of 5 for the respondents from the other three cities.

When the individual items were examined, a large proportion (72.7%) of Thais thought that human immunodeficiency virus (HIV)/acquired immunodeficiency syndrome (AIDS) could be treated with antibiotics. With regard to gonorrhoea, more than 70% of the respondents from Hong Kong and Hangzhou mistakenly answered that the disease could not be treated with antibiotics; while for bladder or urinary tract infection (UTI), the majority of the respondents identified the answer correctly, with the exception being those in Hangzhou, where 53.1% gave the wrong answer. For wound and skin infections, more than 80% of Shanghainese and Hongkongers identified the correct answer, a higher proportion than their counterparts. Please refer to Table 3 and Figure 1 for details.

### 2.4. Knowledge about Antibiotic Resistance

The respondents were asked whether the eight statements about antibiotic resistance were “true” or “false”. Items 2, 3, 4, and 8 were true, while the remaining items were false. A large proportion of respondents from Hong Kong and Shanghai responded correctly to item 2 (“Many infections are becoming increasingly resistant to treatment by antibiotics”) and item 8 (“Antibiotic-resistant infections could make medical procedures like surgery, organ transplants and cancer treatment much more dangerous”). For item 5 (“Antibiotic resistance is an issue in other countries but not here”), 79.4% of the Shanghainese correctly identified this as a false statement compared with a lesser percentage in the other three cities. The median for items that were answered correctly by Shanghainese was 5, with an interquartile range of 4–6, compared to the same median, with an interquartile range of 3–6 for the respondents from Hangzhou, while the Thais and Hongkongers achieved a median of 3 and 4, respectively. Please refer to Table 4 for details.

### 2.5. Predictors of Knowledge about Antibiotic Resistance: Regression Analysis

A multiple linear regression was applied to identify the predictors of knowledge about antibiotic resistance. The potential determinants were based on previous literature about factors that are causing the associations between antibiotics use and antibiotics resistance. The dependent variable was the total number of correct items on antibiotic resistance, while the independent variables included the total scores for knowledge of antibiotic use, city of residence, educational attainment, economic situation, whether to consult doctors on using antibiotics, to consult a doctor before using antibiotics, accommodation, marital status, and gender. All the above variables were included in the regression analysis. A stepwise elimination procedure was used to eliminate the insignificant variables. There were five variables left in the final model. The model showed that respondents who lived in Hong Kong, Shanghai, Hangzhou (reference group: Bangkok), who had higher scores on antibiotics knowledge, a higher level of educational attainment, and who were more likely to consult a doctor on using antibiotics were more likely to give correct answers about antibiotic resistance. Responders who were elderly were less likely to answer the items correctly. The adjusted R^2^ was 0.282. The model was able to explain 28.2% of the variance in the outcome by the significant variables. Please refer to Table 5 for details.

For the city fixed effects regression model, in Hong Kong the predictors include educational attainment, consulting a doctor about using antibiotics, and the score on knowledge of antibiotics. The adjusted R-square is 0.204. Please refer to Table 6.

In Shanghai, the predictors include educational attainment and total score on knowledge of antibiotics. The adjusted R-square is 0.106. Please refer to Table 7.

For Hangzhou, the predictors include age group, total score on knowledge of antibiotics, consulting a doctor about using antibiotics, and educational attainment. The adjusted R-square is 0.308. Please refer to Table 8.

For Bangkok, the predictors include the total score on knowledge of antibiotics, educational attainment, and consulting a doctor about using antibiotics. The adjusted R-square is 0.249. Please refer to Table 9 for details.

## 3. Discussion

Inappropriate use of antibiotics due to inadequate knowledge and inappropriate behaviours are alarming problems in most countries in the world. Antibiotic resistance is an abstract concept for community dwellers, who may have little knowledge about biology, bacteria, and infection compared to healthcare professionals [20]. Our results showed that Hongkongers were using fewer antibiotics in the past six months than those from the other cities; most of the Thais believed incorrectly that antibiotics could be used to treat HIV/AIDS; and that socioeconomic status, knowledge on antibiotics, age, and consulting doctors about the use of antibiotics are predictors of knowledge on antibiotic resistance.

In this study, we classified Shanghai and Hong Kong as high-income cities. According to China Daily, Shanghai was ranked first among cities in China, with the highest per capita GDP in China in 2019 [21], while Hong Kong was ranked 13th out of 50 high-income economies [22]. Hangzhou is a middle-income city and Thailand is a middle-income country. The World Bank upgraded Thailand from a low-middle income economy to an upper-middle income economy in 2021 [23].

Hongkongers used fewer antibiotics in the past six months compared with respondents from the other three cities. According to legislation, in Hong Kong antibiotics can only be obtained from a pharmacy with a doctor’s prescription [24]. In addition, the Hong Kong respondents trusted the doctor’s decision to prescribe antibiotics. [25]. Our study revealed a majority (70.6%) of Thais had brought a child aged 15 years old or below to consult a doctor and asked for antibiotics to treat the common cold. The use of antibiotics by the general public is related to the prescription practices of doctors, public health knowledge, and policies governing the purchase of antibiotics. Although the public are unable to change the prescribing practices of medical practitioners, they should be well informed to not demand antibiotics from the doctor and to follow the doctor’s advice when taking antibiotics. Healthcare professionals are playing an essential role in disseminating information to both hospitalised patients and people in community, to reduce the overuse of antibiotics. Information that antibiotics do not cure colds or flu, and their rational use for children and the young could be emphasised. A recent study in the US [26] suggested using a statement focusing on individual harm such as “antibiotics kill your normal gut bacteria, this can cause bad bacteria to overgrow”, instead of highlighting societal harm such as “antibiotics are the most common cause of Emergency Room visits for drug reactions in children” to educate individuals that they should not request nonindicated antibiotics. 

Regarding specific knowledge on the conditions or diseases that can be treated with antibiotics, other than Hongkongers most respondents incorrectly believed that antibiotics could be used to treat HIV/AIDS. That the Hong Kong respondents were better informed on this point could be related to their higher level of educational attainment, with 19.3% having attained a bachelor’s degree or above. Antibiotics cannot be used to treat HIV/AIDS; however, with prudent use, antibiotics can reduce early mortality in HIV-infected adults and children on antiretroviral therapy [27]. For the treatment of bladder infections, or skin or wound infections, most of the respondents, regardless of whether they were from high or moderate-income cities, were able to provide the correct answers. Most of the respondents and their family members might have caught some of the diseases mentioned, resulting in an increase in knowledge of how the infections were being treated. As for gonorrhoea, it was the disease about which the fewest people answered correctly.

For antibiotic resistance, the median of the correct answers ranged from 3 to 5 among the respondents in the four cities. Unlike knowledge of antibiotics, antibiotic resistance is considered an abstract issue by people from non-healthcare disciplines. A large proportion of the respondents mistakenly identified the following as a true statement: “Antibiotic resistance occurs when your body becomes resistant to antibiotics and they no longer work as well”. According to the World Health Organization, antimicrobial resistance implies that antibiotics become less effective and infections become increasingly difficult or impossible to treat [28]. The large number of incorrect responses could be related to the way the question was phrased, which might have caused a misunderstanding, as the antibiotics could still work but the infection will become more difficult to treat. Another statement, “Bacteria which are resistant to antibiotics can be spread from person to person”, was mistakenly identified by most of the respondents as true. According to the Centers for Disease Control and Prevention, antibiotic resistance affects not only people but can spread between animals and people, making both animals and people sick [29]. The large number of incorrect answers could be related to their being mainly city dwellers, with only 0–3.1% currently serving as agricultural/fishery workers. Although some of the respondents had a high level of educational attainment, they could hardly imagine that bacteria can spread between animals and people through food or contact with animals. The interconnected threat of antibiotic resistance is a risk for individuals and the community, and educational information with a positive attitude to antibiotics is warranted to draw people’s attention to this point. Similarly, national farming policies could mandate that farmers give fewer antibiotics to food-producing animals.

The results of the regression analysis in the present study showed that respondents who lived in Hong Kong, Shanghai, and Hangzhou (reference group: Bangkok), who had a higher level of educational attainment, and who consulted doctor about using antibiotics were likely to give more correct answers, while older respondents were less likely to give correct answers on antibiotic resistance. For the city fixed regression model, educational attainment and knowledge on antibiotic use are predictors among the four cities. The above associations imply that socioeconomic status, living cities, and knowledge on antibiotic use are contributing to knowledge about antibiotic resistance. 

A study conducted in Hong Kong in 2012 [30] showed that a lack of antibiotics knowledge was associated with nonadherence behaviours, which is independent of education. Another study revealed that people who have higher levels of education, who belong to higher wealth quintiles, and who have received antibiotics and antimicrobial information have significantly higher levels of knowledge about the use of antibiotics [31]. Despite the inconsistent results, higher education can be said to influence students’ deep information processing approaches as well as their expectations and values [32]. When educating the public on the proper use of antibiotics and antibiotic resistance, the strategies could be multifaceted and focus on both short and long-term outcomes. The short-term interventions are health education and campaigns for students and the lay community, while the long-term strategies include improving general education targeting people’s knowledge, beliefs, and decision-making skills. Medication adherence is complex and requires a rational decision-making process to achieve the desired behaviours.

Consistent with recent studies conducted in Asia, the US, and Norway [33,34,35], our results show that people with more knowledge of antibiotic use are more likely to give more correct answers about antibiotic resistance. A recent systematic review [36] added empirical information about the public’s knowledge and beliefs and showed that people often have an incomplete understanding of antibiotic resistance and do not believe that they contribute to its development. Interventions to educate the public are encouraged to address the knowledge gap. Other than transmitting knowledge on antibiotic use, factors contributing to antibiotic resistance such as self-medication, nonadherence to medication regimes, neglect of personal hygiene, and psychological barriers that lead to vaccination hesitancy and refusal should be duly considered. Government policies on prescriptions for antibiotics in China and vaccination regime for all age groups in the four cities to raise the awareness about antibiotics and antibiotic resistance.

Older people were less likely to give correct responses on antibiotic resistance. In one Asian country, the majority of the older adults incorrectly believed that antibiotic resistance occurs when the body becomes resistant to antibiotics [37]. About 71.8% of the elderly respondents had never read about or heard of the term antibiotic resistance, although they had a moderate level of knowledge and a positive attitude towards use of antibiotics [38]. Although a number of investigations have been undertaken, there has been no concerted effort made in the four cities targeting older people and grandparents. The education efforts could reduce the incidence of grandparents asking for antibiotics for youngsters during medical consultations.

There are limitations to this study, which should be taken into account. First, although quota sampling was used in the four cities, non-random sampling was used to recruit subjects in each stratum within the cities. Second, the cross-sectional design of the study made it impossible to determine the cause and effect of the study variables. In future studies, representative samples stratified by region should be collected to further analyse antibiotic use and knowledge in moderate and high-income cities.

## 4. Materials and Methods

### 4.1. Study Design and Study Sampling

A quantitative cross-sectional survey was used for the study. The data collection period was from January to August 2021.

The participants were recruited from the urban areas of Shanghai and Hangzhou in China, Bangkok in Thailand, and the Hong Kong Special Administrative Region of China. These four cities were selected because they represent high and middle-income populations in Asia. Quota sampling was used in the study to uphold the representativeness of the study sample. The co-investigators of the respective cities identified the population strata (e.g., major districts in the city), and determined how many participants were to be recruited from each district in the community. Convenience sampling was used in each stratum for subject recruitment. The questionnaires were uploaded to an online survey system. Each computer account could submit the questionnaire only once. The co-investigators of each city were responsible for downloading the questionnaire data. The participants were not compensated for their time in this study.

#### Inclusion and Exclusion Criteria

Persons 18–65 years old, who had been taking medicine and using antibiotics in the past 12 months, who had a job, and who were able to read and communicate in the local language were eligible for participation. People who were cognitively impaired and unable to communicate in the local language were excluded.

### 4.2. Sample Size

It was estimated that a sample size of 601 would be required for this study. This estimate was based on the proportion of the population that was thought to take antibiotics. The standard deviation of the unknown prevalence is required to calculate the size of a sample. As the proportion (p) was unknown, a conservative estimate was set at *p* = 0.50 to produce the most conservative estimate of the sampling error. With a confidence level of 95% and a margin of error of 0.05, a sample size of no less than 385 would be required [39]. For a lower margin of error at 0.04, the required sample size would be 601. To adjust for the possibility that a large amount of data on the respondents would be missing, a sample size of 640 would be required. In total, 160 subjects would be recruited from each study site. Below is the formula that was used to calculate the size of the sample.
(1)$$n=\frac{Z^{2}P(1-P)}{d^{2}}$$ where n = sample size

Z = Z statistic for a level of confidenceP = expected prevalence or proportion (if 20%, P = 0.2)d = precision (in a proportion of one; if 5%, d = 0.05)

### 4.3. Data Collection Method

Data were collected using a questionnaire. The questionnaires were delivered online, due to the COVID-19 pandemic. The questionnaires that were returned were treated as anonymous.

Questionnaire for Data Collection

There were two sections in the questionnaire. The first section examined the participants’ use of antibiotics, and their knowledge of antibiotics and antibiotic resistance. The second section, which was developed by the researchers, was for collecting data on demographic characteristics.

Antibiotic Resistance: Multi-Country Public Awareness Survey

The questionnaire developed by the WHO for conducting a multi-country public awareness survey on antibiotic resistance was adopted in this study. The survey tool was developed by a specialised research agency, in collaboration with the WHO. A field test was carried out by the research agency before data were actually collected [1]. The original version was designed in the English language. A structured bilingual (Cantonese Chinese and English) scale was designed in Hong Kong with additional questions on promoting the safe use of antibiotics and doctor–patient communications [9]. A pilot study consisting of 31 successfully completed interviews was conducted to confirm the validity and reliability of the scale.

The questionnaire used in this study was developed based on the above Cantonese Chinese scale by the Centre for Health Protection, Hong Kong. The scale used in this study was a shorter version of the original questionnaire, incorporating the questions related to the general public’s understanding of the effect of antibiotic use, their attitudes and practices on antibiotic use, and their knowledge and awareness of the problem of antimicrobial resistance. For a test of the content validity test of the scale, five healthcare experts were invited to determine the relevance of the items in the scale. All of the items were considered appropriate for the present study. Cohen’s kappa was calculated for each item to determine the reliability of the scale. Cohen has suggested that inter-rater agreement is substantial if the kappa is between 0.61 and 0.80 [40]. The weighted kappa ranged from 0.64 to 1.00, with a majority of the items being above 0.70, indicating reliability.

The newly developed scale was translated into Simplified Chinese and Thai and underwent validity and reliability tests in China and Bangkok before data were collected. Approvals for the adoption of the scales were obtained from the World Health Organization and Centre for Health Protection, Hong Kong.

Demographic Characteristics

Data that were collected on the demographic characteristics of the respondents included information on their age group, gender, financial situation, level of education, number of household members, and so on.

The study was started in June 2020 and completed in August 2021. It was conducted during the COVID-19 pandemic. Online subject recruitment was used instead of the face-to-face approach. The co-investigators may not be able to explain the questionnaires resulting in some invalid data.

### 4.4. Data Analysis

A statistical analysis was performed using IBM SPSS Statistics for Windows, version 26.0 (IBM Corp, Armonk, NY). Data on demographics, the use of antibiotics, knowledge of antibiotics and antibiotic resistance were put through a descriptive analysis. A one-way analysis of variance (ANOVA) or Kruskal–Wallis test was used to determine the statistically significant differences between the participants in the four cities. Multiple linear regression was used to determine the demographic and social predictors of knowledge of antibiotic resistance. A *p*-value of <0.05 was considered statistically significant.

## 5. Conclusions

The present study showed that people in middle-income cities such as Hangzhou and Bangkok used more antibiotics. Hong Kong people had more knowledge of antibiotics, which could be related to a higher level of educational attainment. On questions about diseases that can be treated with antibiotics, gonorrhoea was the disease that was least likely to receive a correct answer. In general, people with more knowledge about antibiotic use were more likely to give correct answers about antibiotic resistance. Respondents who lived in Hong Kong, Shanghai, and Hangzhou (reference group: Bangkok), who had a higher level of educational attainment, and who were more likely to have consulted a doctor about using antibiotics were more likely to give correct answers on antibiotic resistance, whilst older respondents were less likely to give correct answers on antibiotic resistance. When educating the public on the proper use of antibiotics and antibiotic resistance, the strategies that are employed should be multifaceted and focus on both long-term and short-term outcomes.

## Figures and Tables

**Figure 1 antibiotics-10-01522-f001:**
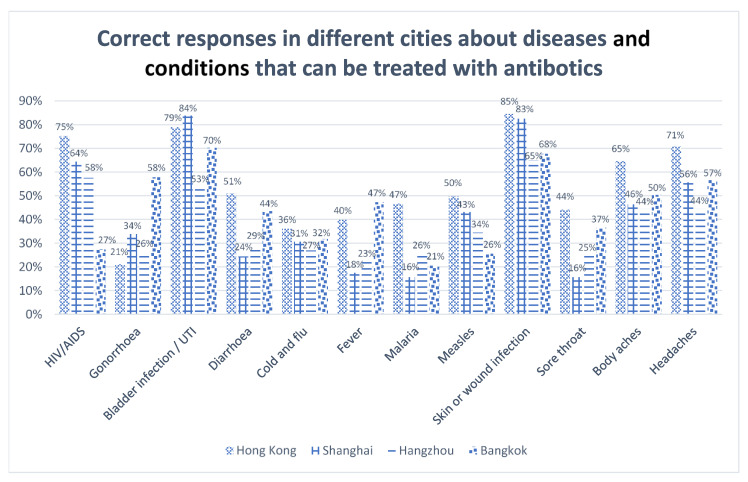
Correct responses about diseases that can be treated with antibiotics.

**Table 1 antibiotics-10-01522-t001:** Demographic characteristics.

	Hong Kong (*n* = 161)	Shanghai (*n* = 160)	Hangzhou (*n* = 160)	Bangkok (*n* = 161)	
	*n* (%)/Median (IQR)	*n* (%)/Median (IQR)	*n* (%)/Median (IQR)	*n* (%)/Median (IQR)	*p*-Value
Gender					* <0.001
Male	74 (46%)	105 (65.6%)	83 (51.9%)	68 (42.2%)	
Female	86 (53.4%)	55 (34.4%)	77 (48.1%)	91 (56.5%)	
Transgender/others	1 (0.6%)	0 (0%)	0 (0%)	2 (1.2%)	
Age					* <0.001
18–30	86 (53.4%)	36 (22.5%)	53 (33.1%)	54 (33.8%)	
31–40	23 (14.3%)	72 (45%)	22 (13.8%)	55 (34.4%)	
41–50	20 (12.4%)	39 (24.4%)	70 (43.8%)	37 (23.1%)	
51–65	32 (19.9%)	13 (8.1%)	15 (9.4%)	14 (8.8%)	
Educational attainment					* <0.001
Primary or below	14 (8.7%)	1 (0.6%)	3 (1.9%)	3 (1.9%)	
Lower secondary	13 (8.1%)	5 (3.1%)	15 (9.4%)	10 (6.2%)	
Upper secondary	31 (19.3%)	17 (10.6%)	36 (22.5%)	39 (24.2%)	
Sub-degree or bachelor’s degree	72 (44.7%)	120 (75%)	78 (48.8%)	92 (57.1%)	
Above bachelor’s degree	31 (19.3%)	17 (10.6%)	28 (17.5%)	17 (10.6%)	
Marital status					* <0.001
Never married	102 (63.4%)	31 (19.4%)	32 (20%)	77 (47.8%)	
Married and with child (ren)	41 (25.5%)	88 (55%)	115 (71.9%)	71 (44.1%)	
Married and without child	9 (5.6%)	30 (18.8%)	11 (6.9%)	11 (6.8%)	
Divorced/Separated/ Widowed	9 (5.6%)	11 (6.8%)	2 (1.3%)	2 (1.2%)	
Occupation					* <0.001
Employer/Manager/ Administrator	8 (5%)	21 (13.1%)	21 (13.1%)	11 (6.8%)	
Professional	46 (28.6%)	62 (38.8%)	38 (23.8%)	34 (21.1%)	
Associate Professional	12 (7.5%)	4 (2.5%)	26 (16.3%)	6 (3.7%)	
Clerk	22 (13.7%)	23 (14.4%)	1 (0.6%)	26 (16.1%)	
Service worker	36 (22.4%)	16 (10%)	30 (18.8%)	34 (21.1%)	
Shop sales worker	14 (8.7%)	9 (5.6%)	17 (10.6%)	11 (6.8%)	
Agricultural/Fishery worker	5 (3.1%)	0 (0%)	1 (0.6%)	4 (2.5%)	
Craft and related worker	4 (2.5%)	3 (1.9%)	9 (5.6%)	1 (0.6%)	
Plant and machine operator and assembler	8 (5%)	6 (3.8%)	1 (0.6%)	4 (2.5%)	
Unskilled worker	6 (3.7%)	4 (2.5%)	13 (8.1%)	4 (2.5%)	
Others	0 (0%)	12 (7.5%)	3 (1.9%)	26 (16.1%)	
Economic situation					* <0.001
Very adequate	6 (3.7%)	2 (1.3%)	0 (0%)	2 (1.2%)	
Adequate	68 (42.2%)	38 (23.8%)	51 (31.9%)	78 (48.4%)	
Barely adequate	60 (37.3%)	67 (41.9%)	81 (50.6%)	51 (31.7%)	
Not adequate	22 (13.7%)	44 (27.5%)	21 (13.1%)	22 (13.7%)	
Very inadequate	5 (3.1%)	9 (5.6%)	7 (4.4%)	8 (5%)	
Religion					* <0.001
Catholic	14 (8.7%)	3 (1.9%)	2 (1.3%)	1 (0.6%)	
Christian	19 (11.8%)	2 (1.3%)	35 (21.9%)	1 (0.6%)	
Buddhist	12 (7.5%)	35 (21.9%)	14 (8.8%)	151 (93.8%)	
Taoist/Muslim	0 (0%)	3 (1.9%)	0 (0%)	2/1 (1.9%)	
Others/No religion	116 (72%)	117 (73.1%)	109 (68.1%)	5 (3.1%)	
Number of household members	3 (2–4)	3 (3–4)	4 (3–5)	3 (2–4)	* <0.001
Number of children	0 (0–0)	0.5 (0–1)	1 (0–1)	0 (0–1)	* <0.001

* *p* < 0.05. Continuous data were analysed using the Kruskal–Wallis H test. Categorical data were analysed using a Pearson Chi-square test or Fisher’s Exact test.

**Table 2 antibiotics-10-01522-t002:** Respondents on the use of antibiotics.

	Hong Kong (*n* = 161)	Shanghai (*n* = 160)	Hangzhou (*n* = 160)	Bangkok (*n* = 161)	
	*n* (%)	*n* (%)	*n* (%)	*n* (%)	*p*-Value
When did you last take antibiotics?					* <0.001
In the last 30 days	10 (6.2%)	24 (15%)	12 (7.5%)	45 (28%)	
In the last 6 months	28 (17.4%)	72 (45%)	77 (48.1%)	61 (37.9%)	
In the last year	38 (23.6%)	46 (28.7%)	71 (44.4%)	20 (12.4%)	
More than a year ago	51 (31.7%)	15 (9.4%)	0 (0%)	0 (0%)	
Never	8 (5%)	2 (1.3%)	0 (0%)	12 (7.5%)	
Cannot remember	26 (16.1%)	1 (0.6%)	0 (0%)	23 (14.3%)	
On that occasion, where did you get the antibiotics?					* 0.003
Hospital or clinic	4 (18.2%)	9 (28.1%)	6 (35.3%)	4 (8.5%)	
Medical store or pharmacy	7 (31.8%)	12 (37.5%)	5 (29.4%)	34 (72.3%)	
The internet	1 (4.5%)	1 (3.1%)	1 (5.9%)	0 (0%)	
Friend or family member	1 (4.5%)	2 (6.3%)	3 (17.6%)	3 (6.4%)	
I had them saved up from a previous time	6 (27.3%)	8 (25%)	1 (5.9%)	6 (12.8%)	
Cannot remember	3 (13.6%)	0 (0%)	1 (5.9%)	0 (0%)	
Do you prefer to consult a doctor who had declared that antibiotics should be used responsibly?					* <0.001
Yes	60 (37.3%)	36 (22.5%)	40 (25%)	138 (85.7%)	
No	65 (40.4%)	73 (45.6%)	74 (46.3%)	13 (8.1%)	
Do not know	36 (22.4%)	51 (31.9%)	46 (28.7%)	10 (6.2%)	
Had you asked for antibiotics for a child below 15 years old (for cold or flu) during the last consultation?					* 0.001
Yes	3 (27.3%)	17 (38.6%)	3 (16.7%)	24 (70.6%)	
No	8 (72.7%)	27 (61.4%)	15 (83.3%)	10 (29.4%)	

* *p* < 0.05. Categorical data were analysed by a Pearson Chi-square test or Fisher’s Exact test.

**Table 3 antibiotics-10-01522-t003:** Knowledge of diseases that can be treated with antibiotics.

	Hong Kong (*n* = 161)	Shanghai (*n* = 160)	Hangzhou (*n* = 160)	Bangkok (*n* = 161)	
	*n* (%)/Median (IQR)	*n* (%)/Median (IQR)	*n* (%)/Median (IQR)	*n* (%)/Median (IQR)	*p*-value
Do you think these conditions can be treated with antibiotics					
HIV/AIDS					* <0.001
Wrong answer	40 (24.8%)	57 (35.6%)	68 (42.5%)	117 (72.7%)	
Correct answer	121 (75.2%)	103 (64.4%)	92 (57.5%)	44 (27.3%)	
Gonorrhoea					* <0.001
Wrong answer	127(78.9%)	106 (66.3%)	118 (73.8%)	68 (42.2%)	
Correct answer	34 (21.1%)	54 (33.8%)	42 (26.3%)	93 (57.8%)	
Bladder infection or urinary tract infection (UTI)					* <0.001
Wrong answer	34 (21.1%)	26 (16.3%)	75 (46.9%)	48 (29.8%)	
Correct answer	127 (78.9%)	134 (83.8%)	85 (53.1%)	113 (70.2%)	
Diarrhoea					* <0.001
Wrong answer	79 (49.1%)	121 (75.6%)	114 (71.3%)	91 (56.5%)	
Correct answer	82 (50.9%)	39 (24.4%)	46 (28.7%)	70 (43.5%)	
Cold and flu					0.366
Wrong answer	103 (64%)	111 (69.4%)	117 (73.1%)	110 (68.3%)	
Correct answer	58 (36%)	49 (30.6%)	43 (26.9%)	51 (31.7%)	
Fever					* <0.001
Wrong answer	97 (60.2%)	132 (82.5%)	124 (77.5%)	85 (52.8%)	
Correct answer	64 (39.8%)	28 (17.5%)	36 (22.5%)	76 (47.2%)	
Malaria					* <0.001
Wrong answer	86 (53.4%)	135 (84.4%)	119 (74.4%)	128 (79.5%)	
Correct answer	75 (46.6%)	25 (15.6%)	41 (25.6%)	33 (20.5%)	
Measles					* <0.001
Wrong answer	81 (50.3%)	91 (56.9%)	105 (65.6%)	120 (74.5%)	
Correct answer	80 (49.7%)	69 (43.1%)	55 (34.4%)	41 (25.5%)	
Skin or wound infection					* <0.001
Wrong answer	25 (15.5%)	28 (17.5%)	56 (35%)	52 (32.3%)	
Correct answer	136 (84.5%)	132 (82.5%)	104 (65%)	109 (67.7%)	
Sore throat					* <0.001
Wrong answer	90 (55.9%)	135 (84.4%)	120 (75%)	102 (63.4%)	
Correct answer	71 (44.1%)	25 (15.6%)	40 (25%)	59 (36.6%)	
Body aches					* 0.001
Wrong answer	57 (35.4%)	86 (53.8%)	89 (55.6%)	80 (49.7%)	
Correct answer	104 (64.6%)	74 (46.3%)	71 (44.4%)	81 (50.3%)	
Headaches					* <0.001
Wrong answer	47 (29.2%)	71 (44.4%)	89 (55.6%)	70 (43.5%)	
Correct answer	114 (70.8%)	89 (55.6%)	71 (44.4%)	91 (56.5%)	
Total correct items	7 (4–9)	5 (3–7)	5 (3–6.75)	5 (3–8)	* <0.001

** p* < 0.001.

**Table 4 antibiotics-10-01522-t004:** Knowledge about antibiotic resistance among residents of four cities.

	Hong Kong (*n* = 161)	Shanghai (*n* = 160)	Hangzhou (*n* = 160)	Bangkok (*n* = 161)	
	*n* (%)/Median (IQR)	*n* (%)/Median (IQR)	*n* (%)/Median (IQR)	*n* (%)/Median (IQR)	*p*-Value
1. Antibiotic resistance occurs when your body becomes resistant to antibiotics and they no longer work as well					* <0.001
Wrong answer	146 (90.7%)	124 (77.5%)	94 (58.8%)	148 (91.9%)	
Correct answer	15 (9.3%)	36 (22.5%)	66 (41.3%)	13 (8.1%)	
2. Many infections are becoming increasingly resistant to treatment by antibiotics					* <0.001
Wrong answer	45 (28%)	34 (21.3%)	52 (32.5%)	74 (46%)	
Correct answer	116 (72%)	126 (78.8%)	108 (67.5%)	87 (54%)	
3. If bacteria are resistant to antibiotics, it can be very difficult or impossible to treat the infections they cause					* <0.001
Wrong answer	49 (30.4%)	60 (37.5%)	78 (48.8%)	91 (56.5%)	
Correct answer	112 (69.6%)	100 (62.5%)	82 (51.2%)	70 (43.5%)	
4. Antibiotic resistance is an issue that could affect me or my family					* 0.01
Wrong answer	72 (44.7%)	47 (29.4%)	68 (42.5%)	73 (45.3%)	
Correct answer	89 (55.3%)	113 (70.6%)	92 (57.5%)	88 (54.7%)	
5. Antibiotic resistance is an issue in other countries but not here					* <0.001
Wrong answer	89 (55.3%)	33 (20.6%)	52 (32.5%)	91 (56.5%)	
Correct answer	72 (44.7%)	127 (79.4%)	108 (67.5%)	70 (43.5%)	
6. Antibiotic resistance is only a problem for people who take antibiotics regularly					* <0.001
Wrong answer	103 (64%)	86 (53.8%)	74 (46.3%)	116 (72%)	
Correct answer	58 (36%)	74 (46.3%)	86 (53.8%)	45 (28%)	
7. Bacteria that are resistant to antibiotics can be spread from person to person					* 0.026
Wrong answer	109 (67.7%)	97 (60.6%)	110 (68.8%)	123 (76.4%)	
Correct answer	52 (32.3%)	63 (39.4%)	50 (31.3%)	38 (23.6%)	
8. Antibiotic-resistant infections could make medical procedures like surgery, organ transplants, and cancer treatment much more dangerous					* <0.001
Wrong answer	47 (29.2%)	34 (21.3%)	68 (42.5%)	88 (54.7%)	
Correct answer	114 (70.8%)	126 (78.8%)	92 (57.5%)	73 (45.3%)	
Total number of correct answers	4 (3–5)	5 (4-6)	5 (3-6)	3 (1-4)	*<0.001

* *p* < 0.05.

**Table 5 antibiotics-10-01522-t005:** Regression analysis for antibiotic resistance.

Variables	Unadjusted β (95% CI of β)	*p*-Value	Adjusted R-Square Contribution
City—Hong Kong (reference group: Bangkok)	0.744 (0.36–1.128)	* <0.001	0.098
City—Shanghai (reference group: Bangkok)	1.65 (1.267–2.032)	* <0.001	
City—Hangzhou (reference group: Bangkok)	1.393 (1.011–1.775)	* <0.001	
Total score for knowledge on antibiotics	0.161 (0.112–0.21)	* <0.001	0.101
Educational attainment	0.46 (0.296–0.625)	* <0.001	0.056
Consult a doctor about using antibiotics (reference group: do not consult a doctor about using antibiotics)	1.102 (0.606–1.598)	* <0.001	0.019
Age	−0.194 (−0.333–−0.055)	* 0.006	0.008

* *p* < 0.05. Adjusted R-square = 0.282.

**Table 6 antibiotics-10-01522-t006:** Regression analysis for antibiotic resistance.

Variables	Unadjusted β (95% CI of β)	*p*-Value	Adjusted R-Square Contribution
Educational attainment	0.453 (0.24–0.666)	<0.001	0.146
Consult a doctor about using antibiotics (reference group: do not consult a doctor about using antibiotics)	0.935 (0.217–1.654)	0.011	0.041
Total score for knowledge on antibiotics	0.081 (0.003–0.16)	0.042	0.017

Adjusted R-square = 0.204.

**Table 7 antibiotics-10-01522-t007:** Regression analysis for antibiotic resistance.

Variables	Unadjusted β (95% CI of β)	*p*-Value	Adjusted R-Square Contribution
Educational attainment	0.669 (0.26–1.077)	0.001	0.073
Total score for knowledge on antibiotics	0.128 (0.032–0.224)	0.009	0.033

Adjusted R-square = 0.106.

**Table 8 antibiotics-10-01522-t008:** Regression analysis for antibiotic resistance.

Variables	Unadjusted β (95% CI of β)	*p*-Value	Adjusted R-Square Contribution
Age	−0.772 (−1.085–−0.459)	<0.001	0.164
Total score for knowledge on antibiotics	0.227 (0.109–0.345)	<0.001	0.101
Consult a doctor about using antibiotics (reference group: do not consult a doctor about using antibiotics)	1.474 (0.256–2.692)	0.018	0.027
Educational attainment	0.394 (0.034–0.755)	0.032	0.016

Adjusted R-square = 0.308.

**Table 9 antibiotics-10-01522-t009:** Regression analysis for antibiotic resistance.

Variables	Unadjusted β (95% CI of β)	*p*-Value	Adjusted R-Square Contribution
Total score for knowledge on antibiotics	0.215 (0.12–0.311)	<0.001	0.197
Educational attainment	0.461 (0.118–0.803)	0.009	0.028
Consult a doctor about using antibiotics (reference group: do not consult a doctor about using antibiotics)	1.126 (0.213–2.039)	0.016	0.024

Adjusted R-square = 0.249.

## Data Availability

The data presented in this survey are available on reasonable request from the corresponding author.

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
