# Peer review of "How Socioeconomic, Health Seeking Behaviours, and Educational Factors Are Affecting the Knowledge and Use of Antibiotics in Four Different Cities in Asia"

_antibiotics, 2021, doi:10.3390/antibiotics10121522_

Round 1

Reviewer 1 Report

Report on “How socioeconomic, health seeking behaviours, and educational factors are affecting the knowledge and use of antibiotics in four different cities in Asia”

The authors investigate the role of the place of residence, education, and other sociodemographic factors on individuals’ knowledge about antibiotics. I enjoyed reading the paper and I am on board with the research questions. However, I see some room for improvement when it comes to the exposition of the analysis.

  • I am not familiar with the literature, so I am unfit to judge whether the contribution to the literature of the paper at hand justifies a publication in this journal. However, the authors could go a long way by comparing the study at hand more carefully with the previous evidence mentioned earlier in the introduction. That is, the authors might want to explicitly point out how they improve upon the key studies they cite.
  • In the sampling description, the authors mention that sampling was conducted in a way to ensure representativeness. However, just looking at the share of men in the Shanghai sample, there seems to be an issue with the sampling. This requires at least some explanation. The authors could also compare their sample means in terms of education, income, marital status, etc., with official statistics.
  • The analysis of the predictive power of sociodemographics requires more explanation. It is not quite clear how the stepwise elimination procedure was conducted. I think that this is an important part of the paper and deserves more attention. I would also like to see more specifications and additional statistics on the predictive power. What is the contribution of each of the factors on the adjusted R^2? Does the significance of some of the factors fade out when controlling for other factors? I would also like to see a specification that compares the potential determinants within a given city, e.g., employing a city fixed effects regression model.
  • How were the potential determinants selected? On a related note: How exactly are factors like `economic situation’ measured?
  • The last sentence of the abstract “When educating the public on the proper use of antibiotics and 23 antibiotic resistance, strategies should be multifaceted and focus on both long-term and short-term 24 outcomes, as well as target different age groups.” seems fairly strong given the descriptive nature of the analysis at hand.

Author Response

The authors investigate the role of the place of residence, education, and other sociodemographic factors on individuals’ knowledge about antibiotics. I enjoyed reading the paper and I am on board with the research questions. However, I see some room for improvement when it comes to the exposition of the analysis.

I am not familiar with the literature, so I am unfit to judge whether the contribution to the literature of the paper at hand justifies a publication in this journal. However, the authors could go a long way by comparing the study at hand more carefully with the previous evidence mentioned earlier in the introduction. That is, the authors might want to explicitly point out how they improve upon the key studies they cite.

Thank you very much for the constructive comments to improve the manuscript.

Reply: The introduction has been shortened to make it explicit and concise. Please refer to line 105-106 on how this study can improve upon the key studies that have been cited.

In the sampling description, the authors mention that sampling was conducted in a way to ensure representativeness. However, just looking at the share of men in the Shanghai sample, there seems to be an issue with the sampling. This requires at least some explanation. The authors could also compare their sample means in terms of education, income, marital status, etc., with official statistics.

Reply: Regarding the sampling method, after the quotas were determined for each city, the co-investigators used non-random sampling to recruit subjects in each strata within the cities. The subjects were not randomly selected resulting an uneven distribution in the age groups. The above has been included in line 379, and in line 362.  Due to small sample size and non-randomly sampling, we did not compare the sample mean in terms of education, income…etc with official statistics.

The analysis of the predictive power of sociodemographics requires more explanation. It is not quite clear how the stepwise elimination procedure was conducted. I think that this is an important part of the paper and deserves more attention. I would also like to see more specifications and additional statistics on the predictive power. What is the contribution of each of the factors on the adjusted R^2? Does the significance of some of the factors fade out when controlling for other factors? I would also like to see a specification that compares the potential determinants within a given city, e.g., employing a city fixed effects regression model.

Reply: The contribution of each of the factors on the adjusted R2 were added to Table 5 giving more specifications and additional statistics on the predictive power. Table 5a to 5c have been added to provide a city fixed effects of the regression model.

How were the potential determinants selected? On a related note: How exactly are factors like `economic situation’ measured?

Reply: The potential determinants were selected based on previous literatures. The explanations were added in line 202-204.

The economic situation is to examine the subjective feeling of the participants about their current economic situation. The above information was added in line 131-132.

The last sentence of the abstract “When educating the public on the proper use of antibiotics and 23 antibiotic resistance, strategies should be multifaceted and focus on both long-term and short-term 24 outcomes, as well as target different age groups.” seems fairly strong given the descriptive nature of the analysis at hand.

Reply: The last sentence in the abstract was revised to make it less strong given the descriptive nature of analysis of the study.

Reviewer 2 Report

The authors have conducted a well-designed and analyzed study re: public knowledge and use of antibiotics in HK, Shanghai, Hangzhou and Bangkok. It is a very well written paper - thank you for the contribution to this important infectious disease and public health topic.

COMMENTS:

  • Would be interested in authors sharing in the Discussions section on next steps following this study? Will there be any policy changes or concerted educational efforts from academia / government from the four cities to increase awareness about antibiotics and antibiotic resistance?
  • Consider discussing why Bangkok was chosen as the reference city considering it from is a completely different country with a different set of cultures, values and beliefs vs. the other 3 cities were somewhat similar since they all belong to China?
  • Line 108: Consider providing an explanation as to why more than 10% of the items were found to be incomplete? Also share which items were left incomplete (so that future surveys can be better designed to prevent this from happening again).
  • Page 4, Table 1: Demographic characteristics under the religion section of the table: why were Taoism and Muslim grouped together? Would be interested in seeing how many participants of each religion were included in the study.
  • Page 6, Table 3 and line 249: Another potential limitation to the question regarding HIV/AIDS is that AIDS is a syndrome that includes bacterial opportunistic infections that can be treated with antibiotics. Consider rewording this to just “HIV” in future studies to avoid confusions for study participants.
  • Page 8, Figure 1: Consider changing title and adding “diseases and conditions” or “diseases and disorders” since body aches and headaches are not diseases.
  • Section 4. Materials and Methods: Consider moving this section before the Results section?
  • Lines 339-345: Would authors be able to share how study participants were recruited? Was it a random sampling to limit biases? Were they recruited from clinics? Were participants compensated for their time?
  • Section 4.3 Data Collection, Line 371: Please share the study timeframe and dates from the beginning to the end of the study. It appears that the study was conducted during the COVID pandemic, were there any confounders that were associated with this study population?

Author Response

Would be interested in authors sharing in the Discussions section on next steps following this study? Will there be any policy changes or concerted educational efforts from academia / government from the four cities to increase awareness about antibiotics and antibiotic resistance?

Thank you for the constructive comments for improving the quality of the manuscript.

Reply: The education efforts and policy changes have been added in line 347-349, line 355-356.

Consider discussing why Bangkok was chosen as the reference city considering it from is a completely different country with a different set of cultures, values and beliefs vs. the other 3 cities were somewhat similar since they all belong to China?

Reply: Bangkok was chosen as a reference city due to its language and beliefs are entirely different from the other 3 cities in Asia. The second part of our study is to examine whether culture is having an impact on use of antibiotics. The above has been added in line 204-206.

Line 108: Consider providing an explanation as to why more than 10% of the items were found to be incomplete? Also share which items were left incomplete (so that future surveys can be better designed to prevent this from happening again).

Reply: Actually, the items were found to be invalid instead of incomplete. The explanations were provided in line 112-113. Those invalid items were related to the participants choosing more than one options or having nil reply. As the invalid items are various in the questionnaires among the 4 cities, they will not be shared in the revised manuscript.

Page 4, Table 1: Demographic characteristics under the religion section of the table: why were Taoism and Muslim grouped together? Would be interested in seeing how many participants of each religion were included in the study.

Reply: The number of Taoism and Muslim have been separated in Table 1.

Page 6, Table 3 and line 249: Another potential limitation to the question regarding HIV/AIDS is that AIDS is a syndrome that includes bacterial opportunistic infections that can be treated with antibiotics. Consider rewording this to just “HIV” in future studies to avoid confusions for study participants.

Reply: Yes.  Since the original questionnaire was developed by the WHO, we only selected the items but did not change the questions to keep this an original version.

Page 8, Figure 1: Consider changing title and adding “diseases and conditions” or “diseases and disorders” since body aches and headaches are not diseases.

Reply: Done and please refer to the highlighted.

Section 4. Materials and Methods: Consider moving this section before the Results section?

Reply: The template is specifically for ‘antibiotics’. We follow the sequence of the sections accordingly.

Lines 339-345: Would authors be able to share how study participants were recruited? Was it a random sampling to limit biases? Were they recruited from clinics? Were participants compensated for their time?

Reply: The new information was added in line 379-383. The participants were not randomly selected and were recruited in the community. There was no compensation for their time.

Section 4.3 Data Collection, Line 371: Please share the study timeframe and dates from the beginning to the end of the study. It appears that the study was conducted during the COVID pandemic, were there any confounders that were associated with this study population?

Reply: The information about the timeframe was added in line 448-451. As online subject recruitment was adopted instead of face-to-face, this could result in the a few invalid data as the researchers were unable to explain the questionnaire to the participants.

Reviewer 3 Report

An interesting work focused on the analysis of bacterial resistance knowledge and application of antibiotics in four Asian cities is presented.

The focus of the manuscript fits into Antibiotics Journal topics.

However, I have some important comments on the text and I consider it necessary to deal with them adequately before final acceptance.

  1. A total of 680 respondents were included in the study. I consider the absence of the methodology for selecting these respondents to be a serious weakness of the manuscript. Further, how were the questionnaires distributed? How the questionnaires were returned for evaluation?
  2. The results show that if there was more than 10% of the missing data in the questionnaire, these were discarded. I'm not a statistician, but this should be explained in more detail. How many questionnaires were discarded? Could it have affected the final results?
  3. The discussion states that the statement "Bacteria which are resistant to antibiotics can be spread from person to person, was mistakenly identified by most of the respondents as true" I do not understand this, it is true, there is a clonal horizontal spread of bacteria. Unfortunately, this statement undermines the credibility of the work presented.
  4. I consider it necessary that the work be thoroughly reworked, the introduction and discussion shortened and more focused on the issue of the article. At the same time, I recommend expanding the author's team to include a microbiologist with a deep knowledge of bacterial resistance issues, who would correct incorrect, resp. inaccurate, microbiological expressions in the text.

Author Response

The focus of the manuscript fits into Antibiotics Journal topics.

However, I have some important comments on the text and I consider it necessary to deal with them adequately before final acceptance.

A total of 680 respondents were included in the study. I consider the absence of the methodology for selecting these respondents to be a serious weakness of the manuscript. Further, how were the questionnaires distributed? How the questionnaires were returned for evaluation?

Thank you for the constructive comments for improving the quality of the manuscript.

Reply: The co-investigators identified the population strata of their respective cities and determined how many participants were to be recruited from each district in the community. Convenience sample was used to recruit subjects in each district. The way how the questionnaires were distributed and return for evaluation have been added in line 379-383. Using non-random sampling has been included in the limitation of the study.

The results show that if there was more than 10% of the missing data in the questionnaire, these were discarded. I'm not a statistician, but this should be explained in more detail. How many questionnaires were discarded? Could it have affected the final results?

Reply: The 10% of the questionnaires were considered invalid, 38 questionnaires were discarded, this has been added and revised in line 109-112. A total of 94.4% of the questionnaires were considered valid. As the missing data is 5.6% , it is considered a low threshold for missing data.

The discussion states that the statement "Bacteria which are resistant to antibiotics can be spread from person to person, was mistakenly identified by most of the respondents as true" I do not understand this, it is true, there is a clonal horizontal spread of bacteria. Unfortunately, this statement undermines the credibility of the work presented.

Reply: This is the original phrase from the questionnaire developed by WHO. My explanation was in the next sentence, citing information from the Centers for Disease Control and Prevention, that antibiotics resistance can spread between animals and people but not simply person to person.

I consider it necessary that the work be thoroughly reworked, the introduction and discussion shortened and more focused on the issue of the article. At the same time, I recommend expanding the author's team to include a microbiologist with a deep knowledge of bacterial resistance issues, who would correct incorrect, resp. inaccurate, microbiological expressions in the text.

Reply: The introduction and discussions were shortened to make them less clumsy and focused on issues of the articles.  Some references were removed as a result.  We are able to consult a  microbiologist for information on bacterial resistance issues, the phrases were amended to ensure accuracy and comprehensible to readers who may not be familiar in this area. Please refer to changes in line 293.

Round 2

Reviewer 3 Report

I would like to thank the authors for the revised version of the manuscript.
I believe that this revision has increased the quality of the article and I can now recommend it for acceptance.